# Murine Mast Cells That Are Deficient in IFNAR-Signaling Respond to Viral Infection by Producing a Large Amount of Inflammatory Cytokines, a Low Level of Reactive Oxygen Species, and a High Rate of Cell Death

**DOI:** 10.3390/ijms241814141

**Published:** 2023-09-15

**Authors:** Yeganeh Mehrani, Jason P. Knapp, Julia E. Kakish, Sophie Tieu, Helia Javadi, Lily Chan, Ashley A. Stegelmeier, Christina Napoleoni, Byram W. Bridle, Khalil Karimi

**Affiliations:** 1Department of Pathobiology, Ontario Veterinary College, University of Guelph, Guelph, ON N1G 2W1, Canada; ymehrani@uoguelph.ca (Y.M.); jknapp03@uoguelph.ca (J.P.K.); jkakish@uoguelph.ca (J.E.K.); stieu@uoguelph.ca (S.T.); lchan12@uoguelph.ca (L.C.); astegelm@uoguelph.ca (A.A.S.); napoleoc@uoguelph.ca (C.N.); 2Department of Clinical Science, School of Veterinary Medicine, Ferdowsi University of Mashhad, Azadi Square, Mashhad 9177948974, Iran; 3Department of Medical Sciences, Schulich School of Medicine & Dentistry, University of Western Ontario, London, ON N6A 3K7, Canada; hjavadi@uwo.ca

**Keywords:** bone-marrow-derived mast cells, vesicular stomatitis virus, type I interferon, reactive oxygen species

## Abstract

Mat cells (MCs) are located in the skin and mucous membranes at points where the body meets the environment. When activated, MCs release inflammatory cytokines, which help the immune system to fight viruses. MCs produce, and have receptors for interferons (IFNs), which belong to a family of cytokines recognized for their antiviral properties. Previously, we reported that MCs produced proinflammatory cytokines in response to a recombinant vesicular stomatitis virus (rVSVΔm51) and that IFNAR signaling was required to down-modulate these responses. Here, we have demonstrated that UV-irradiated rVSVΔm51 did not cause any inflammatory cytokines in either in vitro cultured mouse IFNAR-intact (IFNAR+/+), or in IFNAR-knockout (IFNAR−/−) MCs. However, the non-irradiated virus was able to replicate more effectively in IFNAR−/− MCs and produced a higher level of inflammatory cytokines compared with the IFNAR+/+ MCs. Interestingly, MCs lacking IFNAR expression displayed reduced levels of reactive oxygen species (ROS) compared with IFNAR+/+ MCs. Additionally, upon the viral infection, these IFNAR−/− MCs were found to coexist with many dying cells within the cell population. Based on our findings, IFNAR-intact MCs exhibit a lower rate of rVSVΔm51 infectivity and lower levels of cytokines while demonstrating higher levels of ROS. This suggests that MCs with intact IFNAR signaling may survive viral infections by producing cell-protective ROS mechanisms and are less likely to die than IFNAR−/− cells.

## 1. Introduction

Mast cells (MCs) are essential innate and adaptive immune system components that play crucial roles in allergic and inflammatory responses [1]. They possess unique capabilities contributing to the body’s defense mechanism against viral infections. One of their functions includes degranulation, during which these cells release inflammatory cytokines such as IL-6 and TNF-α [2]. This process aids in initiating and coordinating immune responses against viral invaders. In addition to releasing cytokines, MCs release various mediators that actively participate in the antiviral defense. These mediators include chemokines that attract other immune cells, such as eosinophils and neutrophils, to the infection sites [3]. With the recruitment of these immune cells, MCs promote a coordinated response to the viral infection and enhance the overall effectiveness of the immune system. Despite their pivotal role in combating viral infections, MCs can themselves be susceptible to infection by viruses. Furthermore, MCs have been found to express interferon (IFN) receptors, which are important signaling molecules involved in antiviral defense. The standard host immune response to viral infections involves type I interferon signaling [4]. Type I IFNs are the body’s initial line of defense against viruses. They play a critical role in the host defense system by binding themselves to type I IFN receptors (type I IFNAR), which regulates the innate and adaptive immune response via intracellular antimicrobial programs. Furthermore, they reduce the dissemination of virus particles during the early stages of infection. This allows for the development of a strong adaptive immune response [5,6,7,8,9]. Type I IFN receptors are found in most or all body cells, allowing IFN to effectively protect cells from viral infection [5,10]. In cases of acute viral infection, IFNs are known to exhibit antiviral properties and to limit excessive inflammation by restricting the immune response through downregulating the release of inflammatory cytokines during degranulation and helping MCs to avoid death via immunopathogenesis [8]. INFs also defend against viral infections through activating and boosting antigen presentation in the “early triggered” immune response and triggering the adaptive immune response via direct and indirect action on memory T and B cells [11]. This response is promoted after viral infection since pathogen-associated molecular pattern molecules (PAMPs) signal pathogen recognition receptors (PRRs) through intracellular signaling to stimulate the synthesis and release of interferons via infected cells to ultimately help combat the disease [8]. Interferon signaling might also promote viral defense processes and reduce the vulnerability of MCs on viral replication and reproduction. Our previous study showed that bone marrow-derived mast cells (BMMCs) exposed to recombinant vesicular stomatitis virus (rVSVΔm51) produced IL-6 and TNF-α which was detected with flow cytometry. IFNAR-knockout (IFNAR−/−) BMMCs produced dramatically higher concentrations of the inflammatory cytokines IL-6 and TNF-α than IFNAR-intact (IFNAR+/+) BMMCs. Additionally, cell death was increased upon viral infection in MCs with disrupted type I IFN signaling compared to MCs with BMMCs IFNAR+/+; this was again monitored with flow cytometry [2].

Host antiviral responses are controlled through type I IFNs signaling through IFNAR. Viruses use various strategies to compromise these responses. We demonstrated that blocking IFN responses that were infected with rVSVΔm51 led to an imbalanced cytokine reaction, causing a delay in the elimination of the virus [12]. The protective mechanisms of type I IFN signaling from virus-induced cell death are poorly understood. Apoptotic cells carry out a genetically programmed procedure that results in the ordered disintegration of cellular structures. This tightly regulated process involves various signaling molecules including reactive oxygen species (ROS), and caspases 3 and 7. However, the sources of ROS in MC apoptosis and their role in MC activation have been the subject of conflicting data [13]. It is well known that ROS are deleterious to cells at high levels due to their ability to readily react with proteins, lipids, carbohydrates, and nucleic acids, resulting in programmed cell death [14,15,16]. However, increasing evidence suggests that ROS serve as crucial signaling molecules for regulating hormone action, cell growth, differentiation, apoptosis, transcription, ion transport, immunomodulation, and neuromodulation [16,17,18,19]. ROS also plays a crucial role in the immune defense [16]. As reported in Babior et al. (1973), activated neutrophils and macrophages utilized a membrane-bound NADPH oxidase to generate large amounts of ROS, a process referred to as “oxidative burst”, to destroy viruses and/or microbes and neighboring cells [16,20,21]. The immune system has a number of strategies to contain and eliminate pathogens following pathogen encounters, with ROS being one of them [16]. ROS have also been found to be involved in pathogen detection as well as sensing tissue damage [16]. Professional phagocytes are not the only immune cells to generate ROS; numerous studies have also shown that degranulation of MCs induced via chemical agents or physiological stimuli were also accompanied by ROS generation [13]. We hypothesized that IFNAR signaling might express different proteins during apoptosis in MCs. This study explored the connection between IFNAR signaling and cell infectivity/death and its impact on ROS activity. Murine BMMCs were used as an in vitro model to investigate the relationship between IFNAR and apoptosis. This study aimed to provide data on murine MCs’ response to IFNAR-signaling deficiency to viral infection. These alterations in cytokine production, ROS generation, and cell death rates, help us to understand the immune responses to viral challenges in the presence of INFAR signaling. Understanding the interactions between MCs and interferons in viral immunity can have great implications on developing targeted antiviral therapies and can improve the management of viral diseases.

## 2. Results

### 2.1. Ultraviolet Irradiation of rVSV∆M51 Failed to Induce Cytokines in BMMCs

To explore the role of activation of rVSV∆M51 to induce infection in BMMCs, BMMCs were initially treated with medium, rVSV, and ultraviolet irradiation of rVSV∆M51 at the multiplicity of infection (MOI) of 10 and 50. Ultraviolet irradiation of rVSV∆M51 was used in these experiments to find the necessary level of viral activation to cause infection. Cells were monitored for cytokine generation using bead-based immunoassays (LEGENDplex™ Mouse Inflammation Panel (13-plex), BioLegend, San Diego, United States). The bead array assay showed that irradiated rVSV∆M51 failed to induce the inflammatory cytokines IL-6 and TNFα in BMMCs (Figure 1a,b). This suggests that rVSV∆M51 must be active in order to cause infection in BMMCs, and that the elevation in cytokine production and cell death in BMMCs requires viral replication.

### 2.2. IFNAR Signaling Protected BMMCs from rVSV∆M51-Induced Cell Death

A resazurin assay was used to monitor cell viability/cytotoxicity in rVSVΔm51 infected BMMCs (Figure 2). rVSVΔm51 infected BMMCs exhibited cytotoxicity at the MOI of higher than 10, with IFNAR−/− BMMCs exhibiting the highest cytotoxicity. The metabolic activity of BMMCs IFNAR−/− was reduced significantly at the MOI of 40, indicating that IFNAR signaling protects BMMCs from rVSVM51-induced cell death.

### 2.3. Lack of IFNAR Signaling in BMMCs Affects rVSV∆M51 Infectivity

We used the time-lapse microscopy technique to investigate the possibility of rVSVΔm51 infection in BMMCs. A time-lapse video microscopy study on BMMCs demonstrated that they can be infected with rVSVΔm51, as evidenced by the green fluorescence expressed by the cells. The BMMCs started expressing GFP six hours after adding rVSVΔm51-GFP to the cells, and the GFP expression peaked at 20 h (Figure 3a). Exposure to the virus resulted in a higher number of GFP-positive cells in IFNAR−/− BMMCs compared with IFNAR+/+ BMMCs (Figure 3b), which suggests that IFNAR signaling might play a role in rVSVΔm51 cell infection and death. In this experiment, we utilized the rVSVΔm51 virus, which encodes for a GFP transgene. The GFP protein is only produced when the rVSVΔm51 virus infects the BMMCs, and the GFP transgene is transcribed and translated during virus replication within the BMMCs. Therefore, GFP positivity is indicative of virus infection and does not require any further staining. We then counted and compared the number of GFP-positive cells in each treatment to determine the percentage of infectivity of the BMMCs IFNAR+/+ and IFNAR−/−.

### 2.4. Comparable Viral Titration in IFNAR+/+ and IFNAR−/− BMMCs

In order to identify the factors influencing cell death in BMMCs, a viral plaque assay was conducted. This involved examining the number of plaque-forming units (pfu) and observing the rVSVΔm51 titration per single cell. The results of the viral titer assay showed that there was no variation in viral titration between the IFNAR+/+ and IFNAR−/− BMMC populations (Figure 4). This suggests that the quantity of infectious virus particles generated in both cell culture systems was identical. The data also indicated that the IFNAR−/− BMMCs experienced quick cell death, which helped to balance out their increased susceptibility to rVSVΔm51 virus infection.

### 2.5. ROS Assay Revealed Significant ROS Production in IFNAR+/+ BMMCs

ROS were investigated for their antiviral influence on BMMCs. The DCF-DA cellular ROS assay kit was utilized to assess ROS levels in BMMCs, aiming to understand their antiviral function. The findings indicated that there was a notable rise in ROS production when rVSVΔm51 was exposed to both BMMCs (Figure 5). Interestingly, cells with IFNAR−/− showed lower levels of ROS activity in both the uninfected cells and those infected with the rVSVΔm51 virus. This may suggest that ROS plays a significant part in the cell death that occurs in MCs during antiviral responses.

## 3. Discussion

The current study observed that a lack of IFNAR signaling resulted in increased infection in BMMCs, higher cytokine production, higher rates of virus-induced cell death, and lower ROS production. This study also found that rVSV∆M51 must be activated in order to induce infection in BMMCs. Indeed, rVSV∆M51 that underwent ultraviolet irradiation prior to infection failed to induce antiviral cytokine production in IFNAR+/+ and IFNAR−/− BMMCs. Additionally, IFNAR signaling was found to increase ROS production, which is one of the many strategies the immune system employs to contain and eliminate pathogens. It was observed that IFNAR signaling protected MCs from virus-induced cell death, as was shown through the lower metabolic activity of virus-infected BMMCs IFNAR−/− compared to BMMCs IFNAR+/+.

Our previous study demonstrated that rVSVΔm51 was capable of infecting BMMCs, as was evidenced by the decreasing trend in cell viability in virus-treated cells over the course of infection [2]. While all virus-treated cells displayed a reduction in cell viability following infection with rVSVΔm51, IFNAR−/− BMMCs had the lowest number of viable cells compared to BMMCs IFNAR+/+. Furthermore, our previous study also demonstrated that type I IFN signaling during viral infection helped down-regulate antiviral cytokine production following the infection [2].

The low number of MCs with intact morphology within IFNAR-/- BMMCs after exposure to rVSVΔm51 (Appendix A) suggests a high rate of cell death, which is consistent with the results obtained from flow cytometry analysis and the Rezasourin assay. The protective role of IFNAR signaling against VSV-induced cell death in BMMCs observed in this study align with the findings of another study in which VSV was found to preferentially target type I IFN treatment-resistant bladder cancer cells. The authors treated four human bladder cancer cell lines (RT4, MGH-U3, UM-UC3, and KU-7) with either wild-type VSV or a mutant ∆51M variant in vitro. Both viruses were found to preferentially kill the more aggressive, IFN-nonresponsive cell lines UM-UC3 and KU-7, whereas the IFN-responsive cell lines RT4 and MGH-U3 were less susceptible to infection and virus-induced cell death [22]. Indeed VSV was found to selectively replicate in cells with a defective IFN pathway, whereas it was heavily suppressed in cells with an intact IFN pathway [23].

Following these observations, we looked at whether viral activation was required to cause infection in BMMCs, given that the susceptibility of MCs to viral infection varies between different types of viruses. For example, while human MCs are capable of becoming infected with the influenza A virus, it appears that MCs lack the critical factors essential for productive infection [24,25,26]. Infection of MCs with dengue virus induces MC degranulation, alters cytokine and growth factor profiles, and is found to localize within MC secretory granules [27]. BMMCs, IFNAR+/+, and IFNAR−/− that were treated with irradiated rVSVΔm51 failed to produce the antiviral cytokines IL-6 and TNFα that were observed in BMMCs treated with rVSVΔm51. As such, it is evident that rVSVΔm51 must be activated in order to cause infection and that the elevation in cytokine production and virus-induced cell death was observed in BMMCs. Next, we wanted to elucidate whether IFNAR signaling affected rVSVΔm51 infectivity. Type I interferons are produced early during viral infection [28]. They signal via the alpha/beta interferon receptors to induce the transcription of genes responsible for encoding proteins that limit viral replication and regulate innate and adaptive immune responses [28]. While the protective role of IFNs against viruses is well known, increasing evidence suggests that when type I IFNs are in excess or dysregulated, toxicity, leukopenia, autoimmunity, increased susceptibility to certain infections, and immunosuppression during chronic viral infections can occur [8,29,30]. The ability of IFNAR signaling to mediate protective or detrimental effects in certain circumstances over others depends on the pathogen, the host, and the context [30]. For example, whether the upregulation of the expression of apoptosis-inducing proteins via IFNAR signaling results in tissue damage or immunosuppression during an infection is dependent upon whether these proteins are expressed via non-hematopoietic somatic cells or immune cells, respectively [30]. It has also been argued that the ability of IFNAR signaling to mediate opposing immunological consequences may also depend upon the strength of the IFNAR signaling [29]. According to Shahangian et al. (2009), a strong type I IFN response following infection with influenza sensitized the host to secondary bacterial infections. Indeed, influenza-infected IFNAR−/− mice displayed improved survivability and the clearance of secondary streptococcus pneumonia infection compared with IFNAR+/+ mice [31]. Human MCs play a part in the development and defense against viral diseases by generating type I interferons and chemokines, influencing both the disease’s progression and the immune response [32]. While the study was based on murine MCs, it could imply that human MCs lacking proper IFNAR signaling might also have an increased tendency to generate inflammatory cytokines. It is worth noting that in certain viral infections such as West Nile virus (WNV), DENV, JEV, and ZIKV, type I IFNs suppress viral replication and protect other cells from infection. This study infected the BMMCs with rVSVΔm51 tagged GFP at an MOI of 10. After 20 h, GFP expression in IFNAR−/− BMMCs was greater than IFNAR+/+ BMMCs. This finding is further corroborated with the results obtained from the plaque assays in which no significant difference in viral titration was found between the BMMCs, despite IFNAR−/− BMMCs displaying higher infection rates and increased cell death. It is possible that the lack of a significant difference in viral titration between the BMMCs could be due to rapid cell death in IFNAR−/− BMMCs, thus leaving little time for viral replication to occur.

Lastly, given that in vitro MCs have been reported to produce ROS following degranulation induced via IgE/antigen, we investigated their antiviral function on BMMCs [28]. In this study, IFNAR+/+ BMMCs were found to produce significantly more ROS than IFNAR−/− BMMCs following an infection with rVSVΔm51. ROS are employed by immune cells as a means of host defense against pathogens [33]. While increasing evidence suggests that ROS may act as promoters of infection under certain circumstances, a lack or inhibition of ROS production should be expected to promote infection under normal circumstances [33]. Indeed, BMMCs with disrupted IFNAR signaling produced the least amount of ROS and had higher levels of infection. Studying how other parts of the immune system, like neighboring immune cells and cytokines, interact with IFNAR signaling and ROS production can give a clearer picture of how the immune system fights off viral pathogens with the development of effective antiviral therapies (Figure 6).

Evidence indicates that excessive ROS production harms cell survival, while a moderate oxidative environment aids cell survival and adaptation to the environment [34]. We conducted an experiment where we exposed BMMCs to N-acetylcysteine (NAC), a compound commonly used as an anti-ROS in vitro, to investigate if cell death observed in MCs after being exposed to the rVSVΔm51 was ROS-mediated. Our results show that cell viability significantly increased in both IFNAR+/+ and IFNAR−/− BMMCs that were preincubated with NAC before rVSVΔm51 exposure (as detected with flow cytometry in Appendix A). These findings suggest that NAC protects against cell death, regardless of the differential levels of ROS expression in the BMMCs. NAC has been shown to convert necrotic cells to apoptotic cells and to shield them from cytotoxicity [35,36,37]. It is worth noting that NAC has been proven to increase the cytotoxicity of other compounds using both ROS-dependent and ROS-independent mechanisms [38]. Additionally, it can enhance hypoxic apoptosis through a mechanism that involves the suppression of NFκB transactivation [39]. Further studies are required to dissect NAC protective mechanisms in BMMCs upon rVSVΔm51 exposure.

## 4. Materials and Methods

### 4.1. Mice

Female C57BL/6 mice aged from 6 to 8 weeks were purchased from Charles River Laboratories in Senneville, Quebec. Dr. Laurel Lenz (University of Colorado, Denver, CO, USA) graciously granted her permission for the use of IFNAR−/− mice from the C57BL/6 background, which were kindly provided by Dr. YonghongWan (McMaster University, Hamilton, ON, Canada) and used as bone marrow donors for BMMC development in vitro. At the University of Guelph (Guelph, ON, Canada), all mice were housed in a specifically pathogen-free isolating facility. The use of mice was authorized by the University of Guelph Animal Care Committee (animal utilization protocol #3807) and complied with the standards advised by the Canadian Council on Animal Care.

### 4.2. Virus

Dr. Brian Lichty, McMaster University, kindly provided a recombinant strain of VSV with a deletion at position 51 of the matrix protein (rVSVΔm51). The virus was used in containment level 2 facilities under the approval of the University of Guelph biosafety committee (biohazard permit #A-367-04-19-05).

### 4.3. Generation of BMMCs

BMMCs were generated according to a previously described methods [3]. Flow cytometry detected greater than 99 percent of c-kit-positive cells in the BMMCs generated.

### 4.4. UV-Inactivated VSV-Δm51-hDCT

The infectivity of rVSVΔm51 is shown via ultraviolet irradiation and inactivated virus particles. In vitro, cultured BMMCs (2 × 10^5^ cells) were treated with UV-inactivated VSV-Δm51-hDCT. After 24 h, bead-based immunoassays were used (LEGENDplex™) (BioLegend, San Diego, CA, USA) to quantify multiple cytokines simultaneously in the supernatants.

### 4.5. Cell Viability Assay

A resazurin assay was conducted to quantify the metabolic activity of BMMCs, which correlates with the cell viability. In the 96-well cell culture plate, PBS (Hyclone, Ottawa, ON, Canada) solution was used to fill the outer wells, which evaporated in the incubator. There were control wells containing only media and others having only cells without virus treatment. BMMCs were plated at a density of 2 × 10^5^ cells/well. The BMMCs were treated with nine different MOIs of rVSVΔm51. The cells were treated with the virus at a range of MOI from 10 to 0.0065 (2.5-fold serial dilution) and incubated for 24 h. After 24 h, resazurin sodium salt (Sigma-Aldrich, Oakville, ON, Canada) was added at a final concentration of 0.25 mg/mL, and fluorescence was quantified with a plate reader (excitation wavelength: 535/25 nm; emission wavelength: 590/35 nm) four hours later. Cell viability was calculated as a percentage change relative to untreated control cells after subtracting background fluorescence based on the average of the wells containing media only.

### 4.6. Time-Lapse Images of VSV-Infected BMMCs

BMMCs were infected with rVSVΔm51 expressing a 428 GFP transgene at an MOI of 10 in a 25 mm cell culture petri dish, and real-time images were recorded 429 over the course of 24 h using time-lapse microscopy (LumaScope, Bioimager, Vaughan, ON, Canada). Representative 430 images were extracted from the files that were generated.

### 4.7. Plaque Assay for Quantifying Viral Titers

Vero cells in a 6-well plate at a density of 350,000 cells/well were plated for 24 h prior to running the plaque assay. The next day, the agarose for overlays was prepared through mixing an appropriate amount of agarose with a proper amount of distilled water to achieve the desired 2% agarose concentration. The 2% agarose (Sigma-Aldrich, Oakville, ON, Canada) was put into a microwave for 1–2 min until it dissolved to melt the agarose. The agarose was equilibrated to 56 °C in a water bath before use. To prepare the media for the plaque assay, we produced a mixture of 2× MEM (Thermo Fisher Scientific, Mississauga, ON, Canada ) and cDMEM (Thermo Fisher Scientific, Mississauga, ON, Canada) with 10% FBS (VWR, Pennsylvania, PA, USA), which consisted of one part 2× MEM and two parts cDMEM. The 2× MEM and cEMEM (VWR, Pennsylvania, PA, USA) mixture was equilibrated to 37 °C in a water bath. Then, 10-fold serial dilutions of the virus (virus supernatant) in serum-free DMEM were prepared. The 10^−1^ dilution was produced by mixing 900 μL of SF-DMEM with 100 μL of virus, and then 100 μL of the 10^−1^ dilution was added to 900 μL of SF-DMEM to create the desired dilutions; the usual range for dilution when preparing a full plate is from 10^−5^ to 10^−10^, while for a half plate, it is from 10^−6^ to 10^−8^. After serial dilutions were prepared, we removed all media from wells via aspirating them and immediately adding 500 μL of each appropriate dilution to their respective wells. We carefully performed this task and ensured minimal plate tilting to prevent the cells from drying out. Afterward, the cells were incubated for 1 h at 37 °C with a gentle rocking every 10 min. This prevented the cells from drying out since the volume added to each well was insufficient to cover the monolayer completely. After 1 h of incubation, we removed all media from the wells, starting with the well containing the most diluted sample via aspirating with a pipette. The wells were washed with 1 mL of PBS. For each 6-well plate, 5 mL of 2% agarose with 5 mL 2× MEM and 10 mL cDMEM- FBS were mixed immediately before overlaying the infected cell monolayers. A total of 2.5 to 3 mL of this mixture was added swiftly to each well allowing the overlay to solidify for approximately 10 min at room temperature before the plates were placed in the cell culture incubator. The plates were incubated for 24 h at 37 °C in 5% CO_2_. The plaques were counted after 24 h of incubation.

### 4.8. Reactive Oxygen Species Assay

Intracellular ROS levels were measured in a 96-well plate using the cellular ROS detection assay kit (Abcam—Cellular ROS Assay Kit, Cambridge, UK) based on a fluorogenic dye, DCF-DA ( Abcam, Cambridge, UK), according to the manufacturer’s protocol. Briefly, 100 μg/mL of DCF-DA was added to the BMMCs in each well. The cells were kept at 37 °C prior to loading with DCF-DA to enhance DCF-DA permeability. Then, BMMCs were incubated for 30 min at 37 °C followed by rVSVΔm51 exposure at MOI 10 for 4 h in the presence of DCF-DA. After the 4 h of incubation, the cells were washed twice with PBS containing BSA. Then they were washed another two times with PBS (1×) before being resuspended in FACS buffer for running on a flow cytometry machine, BD FACS-CANTO II (BD, NJ, USA). Quantitative fluorescence analysis of the cells was performed to measure the fluorescence intensity of DCF.

### 4.9. Statistics

GraphPad Prism version 9 (GraphPad Software, San Diego, CA, USA) was used for all graphing and statistical analyses. The graphs show means and standard errors. One- or two-way analysis of variance was used when means of more than two independent groups were subject to comparison across one or more time points respectively. Statistical significance was defined as *p*-values < 0.05.

## 5. Conclusions

According to our study, IFNAR-signaling can affect how murine MCs respond to viral infections. MCs lack IFNAR-signaling, produce inflammatory cytokines, and have lower levels of ROS. This may indicate the activation of a compensatory mechanism or a different signaling pathway in response to the infection. Additionally, IFNAR-deficient MCs have a higher rate of cell death, suggesting that IFNAR-signaling is important for cellular survival during viral infections. On the other hand, MCs with IFNAR-signaling have higher ROS activity, which improves their antiviral activity. These findings suggest that ROS is crucial in promoting a more effective immune response against viral infections in MCs. Further research could investigate IFNAR signaling and the modulation of ROS activity as potential therapeutic targets for viral infections.

## Figures and Tables

**Figure 1 ijms-24-14141-f001:**
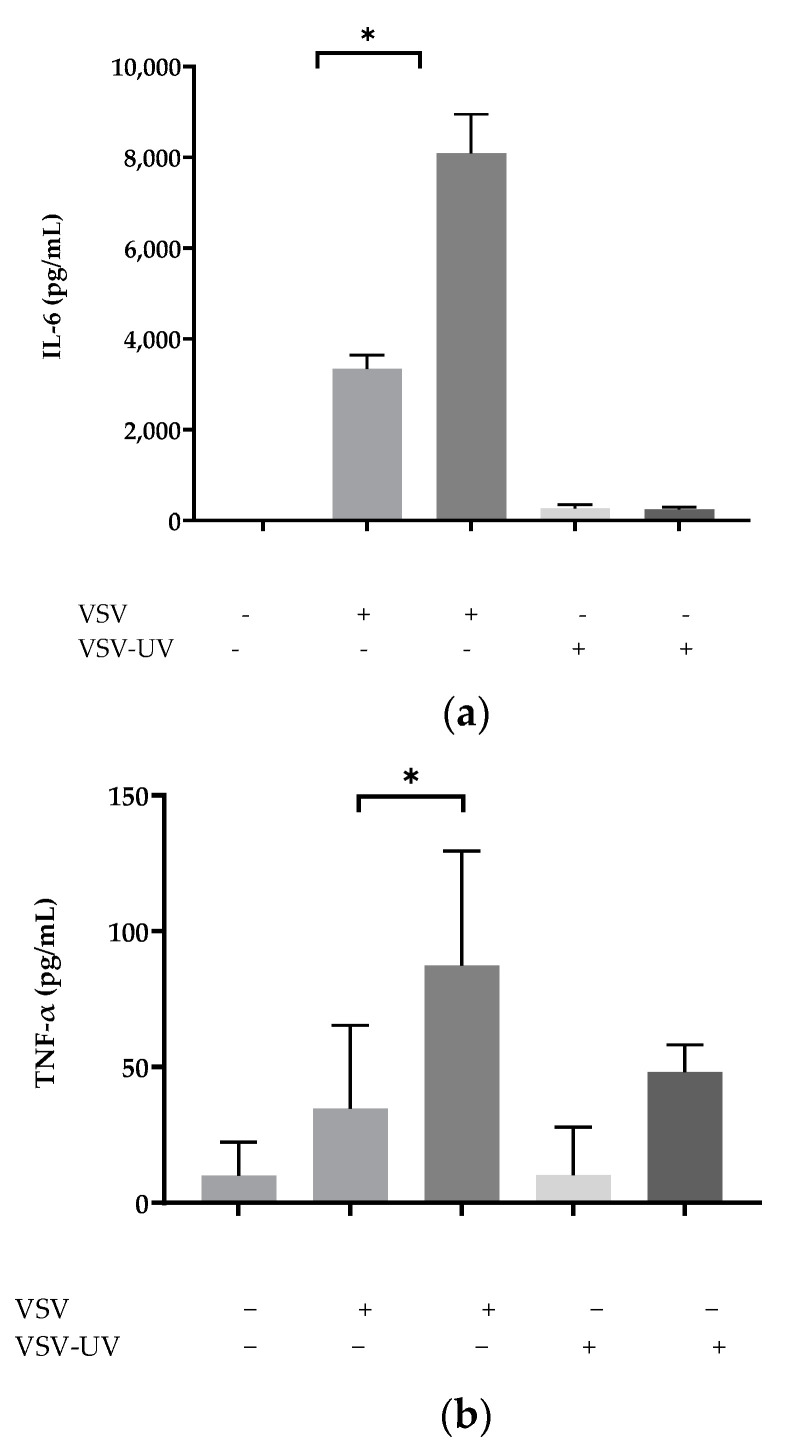
Ultraviolet Irradiation of recombinant vesicular stomatitis virus (rVSVΔm51)Failed to Induce Cytokines in bone marrow-derived mast cells (BMMCs). BMMCs produced IL-6 and TNF-α in response to rVSVΔm51; UV-inactivated rVSVΔm51 does not induce cytokine production. The groups represent control BMMC, infected BMMC with rVSVΔm51 virus at Mot 10 and 50, and infected BMMC with UV-inactivated rVSVΔm51 virus at MoI 10 and 50, respectively. The 24 h post-infection supernatants were assayed for cytokine (IL-6 (**a**) and TNF-α (**b**)) detection using Legend Plex™ kit in. One-way analysis of variance with Tukey’s multiple comparison tests was used to define statistical significance as * *p* < 0.05.

**Figure 2 ijms-24-14141-f002:**
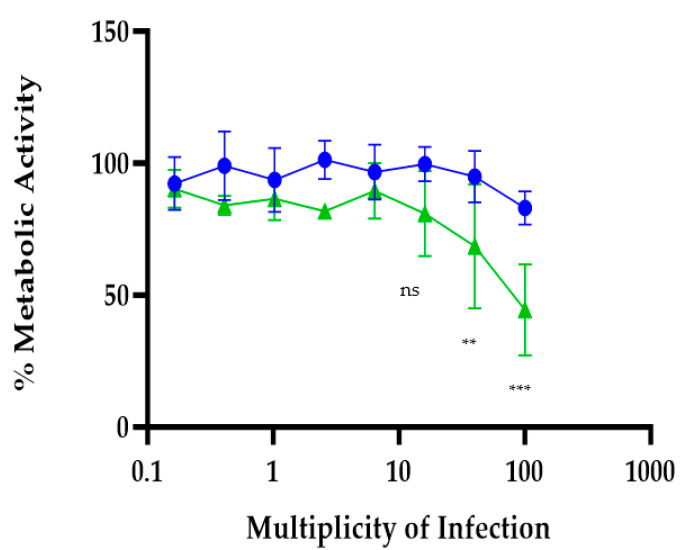
IFNAR Signaling Protected BMMCs from rVSV∆M51-Induced Cell Death. A resazurin assay was conducted to examine the cytotoxicity of rVSVΔm51 in BMMCs. A total of 2 × 10^5^ IFNAR+/+ (blue line/dots) or IFNAR−/− (green line/dots) BMMCs were added to 96-well plates. Then the cells were infected with rVSVΔm51 at different MOIs: 0.1, 0.4, 1, 2.5, 6.4, 16, 40, and 100. The plates were kept in a 37 °C incubator for 24 h. After incubation, the resazurin solution (10% of cell culture volume) was added to the plates and the relative fluorescent intensity was measured after 4 h. Two-way analysis of variance with Tukey’s multiple comparison tests was used to define statistical significance as ** *p* < 0.001, *** *p* < 0.0005, ns *p =* 0.0536.

**Figure 3 ijms-24-14141-f003:**
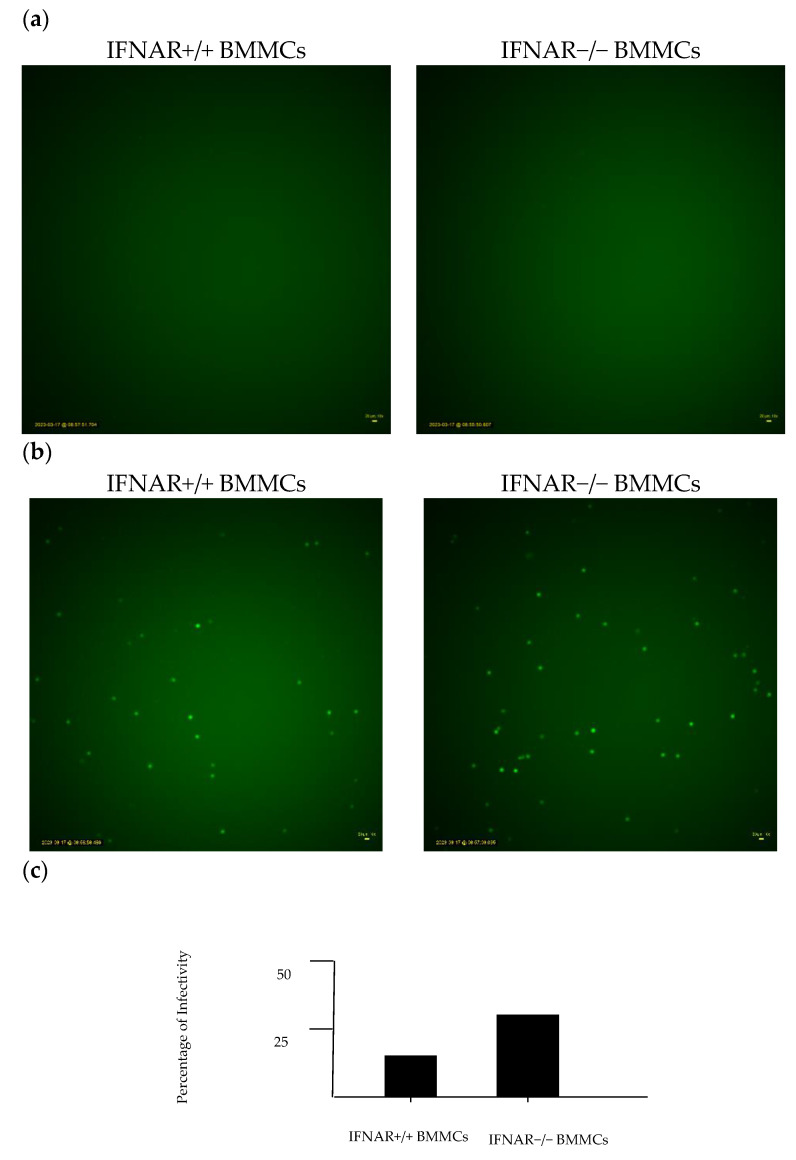
Lack of IFNAR Signaling in BMMCs Affects rVSV∆M51 infectivity. BMMCs were treated with rVSVΔm51 that carried a transgene encoding enhanced GFP at MOI of 10. The time-lapse microscopy technique was then conducted. Fluorescent microscopy of treated BMMCs at (**a**) control 20 h and (**b**) 20 h after exposure to rVSVΔm51-GFP. (**c**) Graphs show the percentage of cells that expressed GFP.

**Figure 4 ijms-24-14141-f004:**
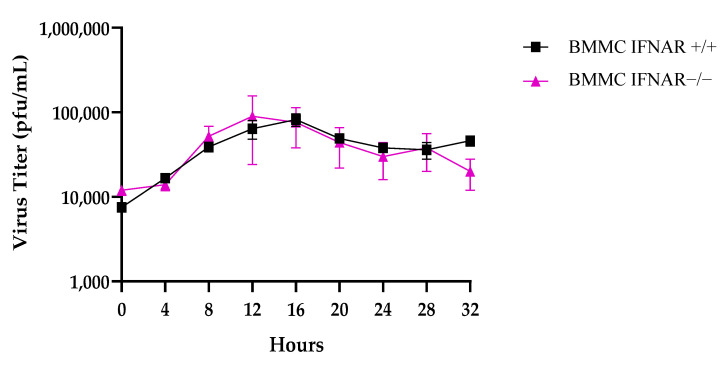
Comparable Viral Titration in IFNAR+/+ and IFNAR−/− BMMCs. BMMCs were infected with rVSVΔm51. The number of plaque-forming units (PFU) was determined using viral plaque assays. The 10^6^ cells/well from IFNAR+/+ or IFNAR−/− BMMCs were infected with an MOI of 10 of rVSVΔm51. After 0, 4, 8, 12, 16, 20, 24, 28 and 32 h post-infection, supernatant was collected for the plaque assay.

**Figure 5 ijms-24-14141-f005:**
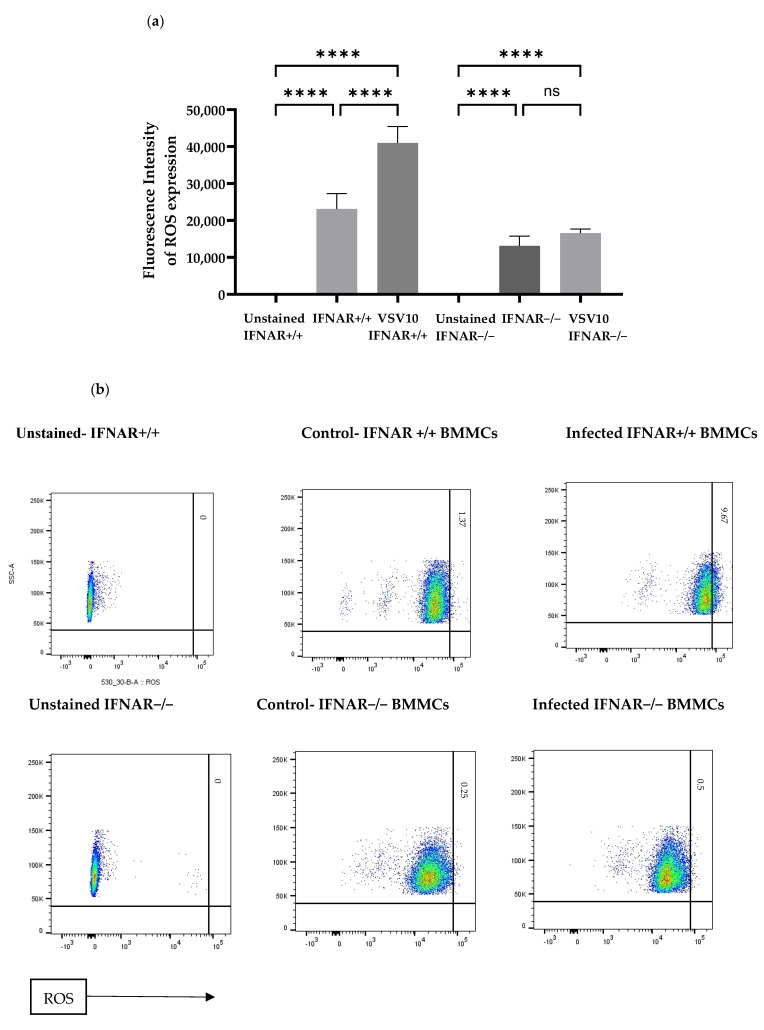
ROS Assay Revealed Significant ROS Production in BMMCs IFNAR+/+. The 2 × 10^5^ BMMCs were left in media or infected with rVSVΔm51 at MOI of 10 for 4 h. ROS activity (the fluorescence intensity of ROS expression) was measured with flow cytometry using a kit as indicated in the Methods and Materials section. Graphs show the fluorescence intensity of ROS expression upon exposure to rVSVΔm51 (**a**), and representative flow cytometry dot plots (**b**) are depicted. One-way analysis of variance with Tukey’s multiple comparison tests was used to define statistical significance as **** *p* < 0.0001.

**Figure 6 ijms-24-14141-f006:**
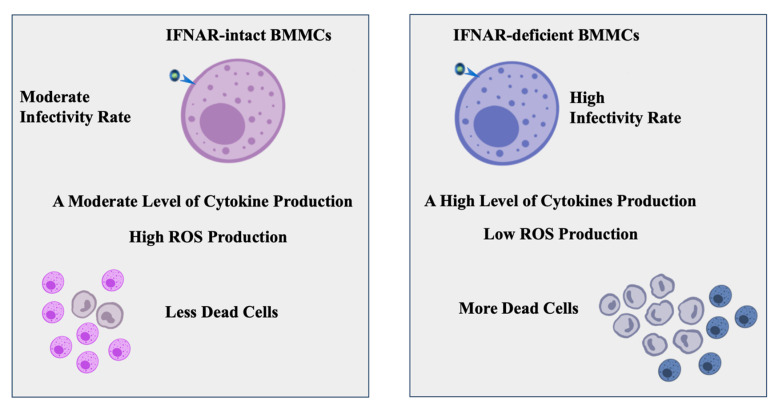
BMMCs with intact IFNAR signaling (IFNA+/+ BMMCs) had moderate sensitivity to viral infection and produced moderate levels of inflammatory cytokines and ROS activity. These BMMCs were also capable of surviving viral infection. On the other hand, BMMCs with disrupted IFNAR signaling (IFNA−/− BMMCs) were highly sensitive to viral infection. They generated high levels of inflammatory cytokines with low levels of ROS activity and exhibited more cellular death during a viral infection.

## Data Availability

Not applicable.

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
