# Peer review of "Murine Mast Cells That Are Deficient in IFNAR-Signaling Respond to Viral Infection by Producing a Large Amount of Inflammatory Cytokines, a Low Level of Reactive Oxygen Species, and a High Rate of Cell Death"

_ijms, 2023, doi:10.3390/ijms241814141_

Round 1

Reviewer 1 Report

Previously, the authors  reported that MCs produced pro- inflammatory cytokines in response to a recombinant vesicular stomatitis virus (rVSVΔm51) and that IFNAR (receptor for IFN) signaling was required to down-modulate these responses. Now the authors  demonstrate that that 30 UV-irradiated rVSVΔm did not cause any inflammatory cytokines in either in vitro cultured  mouse IFNAR-intact (IFNAR+/+) or IFNAR-knockout (IFNAR-/-) MCs, but the non-irradiated virus was able to replicate more effectively in IFNAR-/- MCs and produced a higher level of inflammatory cytokines compared with the IFNAR+/+ MCs. Interestingly, MCs lacking IFNAR expression displayed reduced levels of Reactive Oxygen Species (ROS) compared to IFNAR-intact MCs. Additionally, upon the viral infection, these IFNAR-deficient MCs were found to coexist with many dy ing cells within the cell population. Thae authors propose that IFNAR-intact MCs exhibit a lower rate  of rVSVΔm infectivity and lower levels of cytokines while demonstrating higher levels of ROS. This suggests that MCs with intact IFNAR signaling may survive viral infections by producing cell protective ROS mechanisms and are less likely to die than IFNAR-/- cells.

Very interesting manuscript! I'm curious to see the effects of the treatments on morphology of Mast cells using for example immuhistochemistry. May athe authors show some images of treated mast cells? Are there differences beetween the controls?

Reviewer 2 Report

The authors' article "Murine Mast Cells Deficient in IFNAR-Signaling Respond to Viral Infection by Producing a Large Amount of Inflammatory Cytokines, a Low Level of Reactive Oxygen Species, and a High Rate of Cell Death" is devoted to the study of the mechanisms of resistance of the immune system to viral pathogens. The authors focused on studying the dependence of IFNAR-Signaling on the production of Reactive Oxygen Species, biogenesis of inflammatory cytokines, and apoptosis activity.

The authors showed that BMMCs with impaired IFNAR signaling (IFNA-/- BMMCs) become more susceptible to viral infection. Infected mast cells generated higher levels of inflammatory cytokines with low levels of ROS activity.

The reviewer has clarifying questions

1. Why did the authors conduct the experiment only on Female C57BL/6 mice? Are the results of the study of male mice expected to be identical?

2. How did the authors count the number of cells infected with the virus and thereby determine the efficiency of virus penetration into the cells? Judging by Figure 2, the authors did not use multiplex IHC protocols, so it is difficult to accurately assume the presence of the virus in specific cells.

3. The authors need to explain in more detail the essence of IFNAR Signaling in mast cells under management. In addition, it would be interesting to learn about the proposed changes in other signaling pathways during IFNAR blockade, and how this may affect the state of innate / adaptive immunity during viral infection.

In general, the work is done with an informative design and is interesting. 

Reviewer 3 Report

This work used mouse BMMCs to compare the effect of IFNAR signaling on MC viability and ROS expression following the virus challenge. However, the lack of appropriate control groups in the experiments makes it hard to draw conclusions. Authors previously showed the impact of the virus on BMMC viability and cytokine production and in this work, the progress is to show ROS expression in the last figure, which is not enough for a research manuscript. Not only the control group is missing, no data is provided to discuss the consequence of higher ROS in wild-type BMMCs, and no clear association between ROS, viability, and cytokine production is provided. Below are a few comments:

1.       Fig. 1a:

A-      What is the difference between the two conditions which vsv was added to the cells? The same question about vsv-UV. Please specify in the plots.

B-      Why MOI of 10 is used for the experiments throughout the paper?

2.       Fig. 1b:

A-      Figure legend is missing to specify what the blue and green line/dots represent.

B-      The Resazurin assay is not enough to draw conclusions about cell metabolism. Cell stimulation or infection can change how the cells use different metabolic pathways. While the cells are still alive, they can switch to use a different pathway. This assay can be used to evaluate the viability of cells but for evaluating cell metabolism more specific assays are required.

3.       Fig. 2a:

A-      The pictures should have the same background color, while the picture showing IFNAR+/+ cells is brighter. Not having the same number of cells and background noise is interfering with a fair comparison. Pretty much all the cells in IFNAR+/+ condition are positive for GFP, the only difference is the intensity of GFP expression with some cells showing a lower expression, which is also observed in IFNAR-/- condition. The cells should be stained with DAPI or a dye that stains the nucleus of all the cells to see the cells that are not GFP positive.

B-      A condition that the cells were infected with the virus without GFP (unlabeled virus) needs to be shown and used as a control (some cells might be autofluorescing).

C-      Also, in (b) it is not clear how many cells were considered in each condition. Why not measure GFP by flow cytometry?

4.       Fig. 3:

A-      How did the authors infer that the IFNAR-/- cells experienced quick cell death based on this figure? According to Fig. 1b, at an MOI of 10, there is no significant difference in cell viability between IFNAR+/+ and -/- cells.

5.       Fig. 4:

A-      Line 219 is conflicting with the plot.

B-      The appropriate way is to compare the intensity of expression between stained cells with DCFDA and unstained cells in each group. The unstained control is missing.

C-      The quadrants in IFNAR -/- cells are not matching.

6.       What is the effect of IFNAR deletion in MCs during viral infection in vivo?

7.       What is the implication of this to human mast cells and human viral infection?  

No comment. 
